# Significantly Reduced Alanine Aminotransferase Level Increases All-Cause Mortality Rate in the Elderly after Ischemic Stroke

**DOI:** 10.3390/ijerph18094915

**Published:** 2021-05-05

**Authors:** Sang Joon An, Yun-Jung Yang, Na-mo Jeon, Yeon-Pyo Hong, Yeong In Kim, Doo-Young Kim

**Affiliations:** 1Department of Neurology, Catholic Kwandong University International St Mary’s Hospital, Incheon 22711, Korea; neuroan@gmail.com (S.J.A.); nuyikim@ish.ac.kr (Y.I.K.); 2Institute of Biomedical Science, Catholic Kwandong University International St. Mary’s Hospital, Incheon 22711, Korea; yangyj@ish.ac.kr; 3Department of Rehabilitation Medicine, Catholic Kwandong University International St Mary’s Hospital, Incheon 22711, Korea; 605094@ish.ac.kr; 4Department of Preventive Medicine, College of Medicine, Chung-Ang University, Seoul 06974, Korea; hyp026@cau.ac.kr

**Keywords:** aged, alanine transaminase, brain infarction, frailty, mortality

## Abstract

(1) Background: A significantly reduced alanine aminotransferase (ALT) level is being recognized as a risk factor of increasing mortality in the elderly in relation to frailty. In the elderly, both frailty and ischemic stroke are not only common, but are also associated with mortality. The aim of this research was to investigate whether a significantly reduced ALT level increases the all-cause mortality rate in the elderly with ischemic stroke. (2) Methods: Between February 2014 and April 2019, a retrospective study of 901 patients with ischemic stroke admitted to a university-affiliated hospital was conducted. Cox proportional hazard regression was used to determine whether a significantly reduced ALT level is an independent risk factor for mortality in elderly patients after an ischemic stroke. (3) Results: This study enrolled 323 older adults (age ≥ 65 years) who were first diagnosed with ischemic stroke. The mean age of the participants was 76.5 ± 6.6 years, the mean survival time was 37.1 ± 20.4 months, and the number of deaths was 96 (29.7%). Our results showed that reduced ALT level (less than 10 U/L) increased the risk of all-cause mortality in the elderly after ischemic stroke (adjusted HR: 3.24, 95% CI: 1.95–5.41; *p* < 0.001). (4) Conclusions: A significantly reduced ALT level at the time of diagnosis (less than 10 U/L) is an independent risk factor that increases the mortality rate in the elderly after ischemic stroke.

## 1. Introduction

Serum alanine aminotransferase (ALT) activity has been used as a predictor of overall health status, as well as for liver function monitoring [1]. The liver produces the enzyme ALT; a high serum ALT level is indicative of liver diseases such as cirrhosis or hepatitis. However, recent studies have revealed that a significantly reduced ALT level increases the mortality rate in the elderly population [2,3]. According to a meta-analysis report, each 5 U/L decrease in serum ALT increases mortality [4]. Thus, significantly reduced ALT level could be a result of frailty [2,5]. In elderly individuals with frailty, multiple organ failure may occur. Moreover, sarcopenia may occur because of decreased muscle synthesis, and the synthesis of body-controlled substances may decrease because of decreased liver function. Albumin is synthesized in the liver, and studies have shown an association between a reduced serum albumin level and frailty and mortality [6,7]. ALT is also an enzyme synthesized in the liver that decreases with aging [8].

The term “frailty” refers to a condition of vulnerability to external stress, wherein the physical function is reduced to protect homeostasis; however, frailty itself does not raise the risk of death [9]. Frailty can occur due to illness, but about 32% of elderly individuals without any disease are known to be frail [10]. Frail individuals cannot recover from severe stress situations, which can lead to death [11]. Ischemic stroke is one of the most physically stressful diseases [12]. Patients with stroke experience extreme physical stress, even if they have fully recovered without complications [13].

Frailty and ischemic stroke are common among the elderly. To our knowledge, no study has investigated the relationship between a significantly reduced ALT level and mortality after ischemic stroke. We assumed that a significantly reduced ALT level is related to frailty and could increase the all-cause mortality rate following an ischemic stroke. The aim of this research was to investigate whether a significantly reduced ALT level increases the all-cause mortality rate in the elderly with ischemic stroke.

## 2. Materials and Methods

### 2.1. Participants

Between February 2014 and April 2019, a retrospective study of 901 patients with ischemic stroke admitted to a university-affiliated hospital was conducted. A review of the medical records of these patients diagnosed using magnetic resonance imaging or computed tomography was performed. The exclusion criteria and number of patients excluded were as follows: suspected liver disease (ALT level >40 U/L; n = 92), not the first episode of acute ischemic stroke (n = 244), insufficient medical records (n = 8), and age < 65 years (n =234).

Finally, 323 participants were enrolled in the study (Figure 1), which was approved by the university-affiliated hospital’s Institutional Review Board.

Of the 901 patients, 92 patients with ALT levels over 40 were excluded and 244 patients had a previous history of stroke. Of the remaining stroke patients, 8 patients had insufficient medical records, and 234 patients under the age of 65 were excluded. Finally, 323 subjects were enrolled in the study.

### 2.2. Data Collection

Age, sex, and body mass index (BMI) data were evaluated. The investigated medical data related to ischemic stroke included lesions of ischemic stroke, thrombolytic treatment (including IVtPA (intravenous tissue plasminogen activator) and stent retriever), and etiology, as per the Trial of ORG 10172 in Acute Stroke Treatment (TOAST) classification and history of atrial fibrillation, diabetes, hypertension, coronary artery occlusive disease, cancer, alcohol, and smoking. The initial National Institutes of Health Stroke Scale (NIHSS) score at admission was used to adjust ischemic stroke severity. Laboratory data of albumin, ALT, creatinine (Cr), erythrocyte sedimentation rate (ESR), neutrophil-lymphocyte ration (NLR), glucose, total cholesterol, and hemoglobin (Hb) were collected from the time of diagnosis. Systolic blood pressure (SBP) at admission and systolic blood pressure variability (SPBV) during first 24 h were also investigated. Mortality surveillance was investigated using the National Health Insurance database, as of 28 September 2020.

### 2.3. Data Preprocessing

Factors known to be linearly associated with mortality, such as initial NIHSS score at admission, age, Cr, total cholesterol, and ESR, were used as continuous variables [14,15,16,17,18,19,20,21].

A serum albumin level of less than 3.5 g/dL, glucose level of more than 7.3 or less than 3.7 mmol/L, and Hb level of less than 11 g/dL are all considered risk factors for death [6,18,22]. Both underweight and obesity are considered to be related to a higher risk of death. BMI (kg/m^2^) was used as a categorical measure, and participants were divided into the following four groups: underweight (18.5), average weight (18.5–24.9), overweight (25.0–29.9), and obese (over 30) [23,24]. According to a meta-analysis report, each 5 U/L decrease in serum ALT increased mortality [4]. The criterion for significantly reduced ALT level was set at 10 U/L, based on previous research.

### 2.4. Statistical Analyses

Participants of the study were divided into the expired group (those patients with death after acute cerebral infarction) and surviving group (those patients who survived after acute cerebral infarction). The differences between groups were compared statistically. An independent *t*-test for continuous variables and Fisher’s exact test or chi-square test with a post-hoc Bonferroni test for categorical variables were used. Continuous variables were defined as the mean standard deviation for normally distributed variables and categorical variables as frequency and percentage.

The cumulative risk of overall survival at 79 months was estimated using the Kaplan–Meier method. Stratified analyses were made for the ALT level (<10 U/L, and 10–40 U/L). Time-to-event comparisons were made using the log-rank test.

The relationship between a significantly reduced ALT level (less than 10 U/L) and mortality was assessed using a multivariate Cox proportional model (Forward stepwise: Conditional) that controlled for possible confounding factors including age, sex, TOAST classification, stroke location, hemispheric localization, initial NIHSS at admission, SBP at admission, SBPV, thrombolytic treatment, reduced albumin level (less than 3.5 g/dL), reduced Hb level (less than 11 g/dL), abnormal glucose level (more than 7.3 or less than 3.7 mmol/L), serum Cr, ESR, NLR, total cholesterol levels, BMI, history of hypertension, diabetes mellitus, atrial fibrillation, coronary artery occlusive disease, cancer, smoking, and alcohol consumption. A statistically significant value was defined as a p-value of less than 0.05.

For testing the proportional hazard assumption in the Cox model, first we used a visual inspection of the log minus log curve. Second, a time-dependent Cox regression analysis was additionally performed to investigate whether the time-dependent covariate was not significant.

The statistical analysis was performed using the Statistical Package for the Social Sciences, version 22.0 (IBM Corp., Armonk, NY, USA).

## 3. Results

This study enrolled 323 older adults aged ≥65 years. Their mean age was 76.5 ± 6.6 years and their mean survival time was 37.1 ± 20.4 months; death was reported in 96 (29.7%) cases. IVtPA treatment was performed in 13 patients (4.0%), and the mean time to IVtPA treatment was 68.8 ± 30.5 min. Stent retriever treatment was performed in 20 patients (6.2%), and the mean time to stent retriever treatment was 132.7 ± 60.0 min. IVtPA with stent retriever treatment was performed in three patients (0.9%). The general characteristics of the expired and surviving groups are shown in Table 1. Compared with the surviving group, the expired group was older and normal weight, with higher levels of initial NIHSS at admission, SBPV during the first 24 h, creatinine levels, and ESR value, and with lower levels of ALT, albumin, and Hb (Table 1).

Figure 2 is a Kaplan–Meier survival function with a stratified analysis for the ALT level (<10 U/L and 10-40 U/L). The cumulative 79-month survival was 27.4% in patients with a significantly reduced ALT level versus 63.1% in patients with a normal level of ALT (*p* < 0.001).

The association between the selected risk factors and all-cause mortality was assessed through univariate analysis (Appendix A). Table 2 shows the unadjusted and adjusted hazard ratio (HR) for all-cause mortality for a significantly reduced ALT level. Elderly patients with ischemic stroke with a significantly reduced ALT level had an increased risk of all-cause mortality in both the unadjusted (HR = 2.73, 95% CI 150 1.50–4.98) and adjusted analyses (3.24, 1.95–5.41). In addition, age, initial NIHSS score, and serum Cr levels at diagnosis were also significant risk factors. A visual inspection of the log minus log function showed that the assumption of proportional hazards was appropriate. To further the testing of the proportional hazard assumption in the Cox models, we additionally generated the time dependent covariates by creating an interaction between the ALT level and a function of survival time, and included this in the model. The time dependent covariate was not significant.

## 4. Discussion

The results of this study showed that the factors that increase the mortality rate in the elderly after ischemic stroke were age, initial NIHSS score at admission (indicator of stroke severity), high Cr levels, and a significantly reduced ALT level (less than 10 U/L). It is noteworthy that laboratory tests at diagnosis have a clear correlation with mortality, even though statistically controlled confounding factors are known to be risk factors. Survival analysis using the multivariate Cox proportional hazard analysis revealed that elderly individuals with a significantly reduced ALT level (less than 10 U/L) had a significantly higher mortality rate after ischemic stroke; this difference in mortality rate was observed shortly after the onset and increased over time.

The Totaled Health Risks in Vascular Events (THRIVE) and Acute Stroke Registry and Analysis of Lausanne (ASTRAL) are models for predicting mortality outcomes using acute ischemic stroke states [25]. THRIVE is a tool for predicting mortality and prognosis using NIHSS, age, and other risk factors (hypertension, diabetes, and hyperlipidemia). ASTRAL is also a tool for predicting mortality with NIHSS, age, and abnormal glucose level at admission [18]. Similar to these well-known mortality prediction tools, age and initial NIHSS score were the major mortality predictors in our study. It is obvious that stroke severity and age are related to mortality. The significant association of the initial NIHSS score at admission with stroke mortality suggests that it is not only an indicator of stroke severity, but also a strong predictor of ischemic stroke mortality.

The TOAST classification and tentorial location were not associated with mortality. However, in hemispheric localization, the proportion of the bilateral injury group was higher in the expired group; in the univariate analysis, the unilateral injury group had a lower mortality rate than the bilateral injury group. This could be attributed to the difference in severity according to the extent of the infarction area.

The post-hoc test results for the four BMI categories in our study showed a significant difference in mortality between the overweight and underweight groups; there were more expired cases in the latter. Previous studies have shown that BMI and mortality are inversely related. Elderly individuals with a higher BMI had a lower mortality rate than those with an underweight BMI, which was interpreted as sarcopenia. Thus, the mortality risk due to frailty in patients who are underweight is higher than that in those who are overweight [26,27].

Thrombolytic treatment includes IVtPA treatment and stent retriever treatment. Only 9.3% of the subjects included in this study were included. It is known that the survival rate of thrombolytic treatment may vary depending on the treatment time, but in this study, the sample size was too small for a subgroup analysis, so it could be concluded [28,29].

Recent evidence suggests that high blood pressure and blood pressure variability are important factors in predicting the prognosis of acute stroke. In particular, it is known that systolic blood pressure at admission and systolic blood pressure variability increase the risk of cerebral hemorrhage or symptomatic hemorrhagic transformation, and adversely affect mortality and functional outcome [30,31,32]. As a result of our study, in the comparison between the expired group and the surviving group, there was no difference in SBP at admission, but in the case of SBPV, there was a significant difference between the two groups. However, there was no statistically significant difference when observing the long-term effect. This is thought to be because the majority of subjects included in this study did not progress to cerebral hemorrhage or hemorrhagic transformation.

There is strong evidence that stroke outcomes are strongly influenced by various factors related to metabolic homogeneity, inflammatory reactions, and perfusion disorders [33,34,35,36]. It is known that glycosylated hemoglobin or serum glucose levels at admission are associated with metabolic homeostasis in the prognosis of patients with acute ischemic stroke. In this study, there were many missing data on glycosylated hemoglobin due to the characteristics of the retrospective studies, so glucose at admission was used as a metabolic factor. However, in this study, the long-term effect of glucose levels on mortality was not significant.

In our study, the laboratory findings of the two groups showed that the proportion of a significantly reduced ALT level (less than 10 U/L), reduced albumin level (less than 3.5 g/dL), and reduced Hb level (less than 11 g/dL) was significantly higher in the expired group than in the surviving group. Furthermore, Cr, ESR levels, and NLR were significantly higher in the expired group. Albumin and Hb levels reflect the whole-body nutritional status, high ESR and NLR indicates systemic inflammatory response, and high Cr indicates renal dysfunction. In previous research, all of these factors were found to be strongly related to mortality. For the prognosis of inflammatory response and stroke, ficolin-1 is also a strong prognostic, but it was not included in this retrospective study [3,6,7,19,21,22,35].

Significantly reduced serum ALT is an independent risk factor for mortality in the elderly population [2,3,4,6,7,8]. The mortality rate increased for each 5 U/L decrease in serum ALT, according to a meta-analysis. [4]. A significantly reduced ALT level has been related to frailty in previous studies. The hepatic organ degenerates in fragile elderly people, decreasing the ALT activity in the liver and resulting in a significantly reduced serum concentration [2,37]. In this study, we constructed a Cox proportional model, depending on the level of ALT (10 U/L) and controlling for confounding factors, and found significant differences in elderly mortality after ischemic stroke. These findings are consistent with previous studies and have identified significantly reduced ALT level (less than 10 U/L) as an independent risk factor for all-cause mortality in the elderly after ischemic stroke. Notably, the laboratory results at diagnosis showed significant associations with mortality rate, even though other well-known risk factors were adjusted.

The accumulation of damage in the body’s organ system is the cause of death from frailty. Most people who are afflicted with diseases will recover without dying. Frail people, on the other hand, are more vulnerable to other illnesses and take longer to recover, and their organ damage continues to expand over time, ultimately leading to death. [38]. Rehabilitation can be helpful to people with frailty. There are clear theoretical reasons for using effective rehabilitation approach for frailty, and some studies have suggested its usefulness [20]. Therefore, if ALT levels in elderly patients with acute ischemic stroke are significantly reduced (less than 10 U/L), they are at a high risk of mortality associated with frailty; hence, an active rehabilitation approach for frailty would be required.

This study had some limitations. As the cause of mortality was not investigated, it is not possible to determine whether it was the cause of the ischemic stroke. Therefore, when referring to mortality, it is considered as mortality caused by the all-causative cause. As this study was a retrospective study, information on the modified Rankin scale or NIHSS at discharge was difficult to control and was not included. In addition, as previously mentioned, ficolin-1, which is associated with inflammatory reactions, was not a test widely distributed in the general clinical sites, and thus could not be investigated because of limitations in retrospective design. Although a clear mechanism has not been identified yet, if a prospective design study including compelling evidence related to the prognosis of stroke is conducted, it will be helpful to improve the algorithm for the prognosis of ischemic cerebral infarction in the future. In addition, a limitation of this study is that the subgroup analysis for thrombolytic treatment was not progressed because of the small sample size. In the future, it will be necessary to use cohort studies rather than cross-sectional studies to determine the association between a significantly reduced ALT level, frailty, and mortality. In order to evaluate the efficacy of rehabilitation in patients with a significantly reduced ALT level, further research is required.

## 5. Conclusions

In conclusion, a significantly reduced ALT level at diagnosis (less than 10 U/L) is an independent risk factor that increases the mortality rate in the elderly after ischemic stroke.

## Figures and Tables

**Figure 1 ijerph-18-04915-f001:**
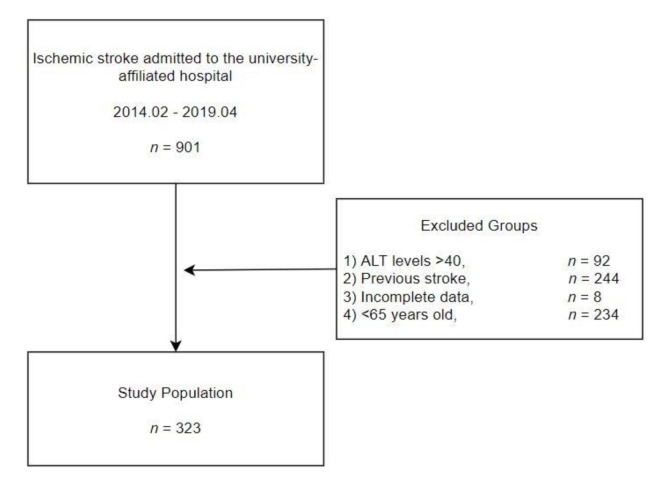
Study population.

**Figure 2 ijerph-18-04915-f002:**
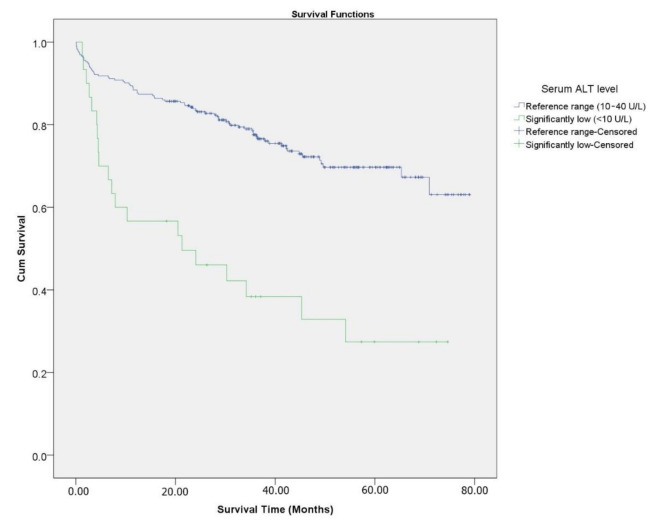
Kaplan–Meier survival function of patients after ischemic stroke.

**Table 1 ijerph-18-04915-t001:** General characteristics and comparison between expired and surviving groups.

Covariates	Total (*n* = 323)	Expired (*n* = 96)	Surviving (*n* = 227)	*p* Value
Age (years)	76.5 ± 6.6	79.3 ± 6.9	75.3 ± 6.1	<0.001 *
Sex, n (%)				0.314
Female	167 (51.7%)	45 (46.9%)	122 (53.7%)	
Male	156 (48.3%)	51 (53.1%)	105 (46.3%)	
Initial NIHSS at admission	5.4 ± 6.5	8.7 ± 8.4	3.9 ± 4.8	<0.001 *
IVtPA treatment, n (%)	13 (4.0%)	5 (5.2%)	8 (3.5%)	0.538
Stent Retriever treatment, n (%)	20 (6.2%)	6 (6.2%)	14 (6.2%)	1.000
IVtPA with Stent Retriever treatment	3 (0.9%)	1 (1.0%)	2 (0.9%)	1.000
Time to IVtPA (minute)	68.8 ± 30.5	76.8 ± 16.9	63.9 ± 38.7	0.500
Time to Stent Retriever (minute)	132.7 ± 60.0	141.3 ± 78.1	129.0 ± 56.0	0.693
SBP at admission	144.2 ± 24.0	145.5 ± 25.4	143.6 ± 23.5	0.529
SBPV during first 24 h	42.3 ± 17.3	45.5 ± 18.7	40.9 ± 16.5	0.031 *
Survival Time (month)	37.1 ± 20.4	18.4 ± 17.0	45.0 ± 16.2	<0.001 *
TOAST, n (%)				0.643
Large-artery atherosclerosis	162 (50.2%)	53 (55.2%)	109 (48.0%)	
Cardioembolism	85 (26.3%)	25 (26.0%)	60 (26.4%)	
Small-vessel occlusion	45 (13.9%)	10 (10.4%)	35 (15.4%)	
Stroke of other determined etiology	1 (0.3%)	0 (0.0%)	1 (0.4%)	
Stroke of undetermined etiology	30 (9.3%)	8 (8.3%)	22 (9.7%)	
Location, n (%)				0.197
Both	23 (7.1%)	8 (8.3%)	15 (6.6%)	
Infratentorial	67 (20.8%)	14 (14.6%)	53 (23.3%)	
Supratentorial	233 (72.1%)	74 (77.1%)	159 (70.0%)	
Hemispheric localization, n (%)				0.036 *
Both	57 (17.7%)	25 (26.0%)	32 (14.1%)	
Left	129 (39.9%)	34 (35.4%)	95 (41.9%)	
Right	137 (42.4%)	37 (38.5%)	100 (44.1%)	
Laboratory findings at diagnosis				
Extremely low ALT (<10 U/L), n (%)	30 (9.3%)	20 (20.8%)	10 (4.4%)	<0.001 *
Low albumin (<3.5 g/dL), n (%)	60 (18.6%)	32 (33.3%)	28 (12.3%)	<0.001 *
Low Hb (<11 g/dL), n (%)	64 (19.8%)	33 (34.4%)	31 (13.7%)	<0.001 *
Random Glucose, n (%)				0.298
Hyperglycemia (>7.3 mmol/L)	72 (22.3%)	22 (22.9%)	50 (22.0%)	
Hypoglycemia (<3.7 mmol/L)	1 (0.3%)	1 (1.0%)	0 (0.0%)	
Reference (3.7~7.3 mmol/L)	250 (77.4%)	73 (76.0%)	177 (78.0%)	
Creatinine (mg/dL)	0.8 ± 0.5	1.0 ± 0.8	0.8 ± 0.3	0.048 *
ESR (mm/h)	23.1 ± 20.4	27.2 ± 22.2	21.3 ± 19.5	0.017 *
NLR	3.5 ± 3.6	4.0 ± 4.5	3.3 ± 3.2	0.094
Total cholesterol (mg/dL)	174.4 ± 42.3	170.6 ± 41.8	176.0 ± 42.7	0.289
BMI, n (%)				<0.001 *
Reference (18.5–24.9)	196 (60.7%)	65 (67.7%)	131 (57.7%)	
Underweight (<18.5)	16 (4.9%)	10 (10.4%)	6 (2.6%)	
Overweight (25.0–29.9)	99 (30.7)	16 (16.7%)	83 (36.6%)	
Obese (≥30)	12 (3.7%)	5 (5.2%)	7 (3.1%)	
Atrial fibrillation, n (%)	82 (25.4%)	24 (25.0%)	58 (25.6%)	1.000
DM, n (%)	103 (31.9%)	32 (33.3%)	71 (31.3%)	0.817
Hypertension, n (%)	227 (70.3%)	64 (66.7%)	163 (71.8%)	0.429
Coronary artery occlusive disease, n (%)	46 (14.2%)	15 (15.6%)	31 (13.7%)	0.773
Cancer history, n (%)	14 (4.3%)	6 (6.2%)	8 (3.5%)	0.423
Smoking history, n (%)	53 (16.4%)	17 (17.7%)	36 (15.9%)	0.806
Alcohol consumption, n (%)	53 (16.4%)	14 (14.6%)	39 (17.2%)	0.681

Values indicate mean ± standard deviation; IVtPA—intrvenous tissue plasminogen activator; ALT—alanine aminotransferase; BMI—body mass index; DM—diabetes mellitus; ESR—erythrocyte sedimentation rate; Hb—hemoglobin; NIHSS—National Institute of Health Stroke Scale; NLR—neutrophil-lymphocyte ratio; SBP—systolic blood pressure; SBPV—systolic blood pressure variability; TOAST—Trial of ORG 10172 in Acute Stroke Treatment; * *p* < 0.05.

**Table 2 ijerph-18-04915-t002:** Unadjusted and adjusted hazard ratios and 95% CI for all-cause mortality, according to the ALT level.

Covariates	Unadjusted HR	(95% CI)	Adjusted HR	(95% CI)
10–40 U/L ALT level	1.00	(Reference)	1.00	(Reference)
<10 U/L ALT level	2.73	(1.50–4.98)	3.24	(1.95–5.41)
Age (years)			1.08	(1.04–1.11)
Initial NIHSS at admission			1.10	(1.07–1.13)
Creatinine (mg/dL)			1.50	(1.16–1.94)

## Data Availability

Not applicable.

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
