# Peer review of "Significantly Reduced Alanine Aminotransferase Level Increases All-Cause Mortality Rate in the Elderly after Ischemic Stroke"

_ijerph, 2021, doi:10.3390/ijerph18094915_

Round 1
Reviewer 1 Report
This was a retrospective study aimed to the association between reduced ALT level and mortality in elderly patients with ischemic stroke. The study enrolled 323 patients. Significantly reduced ALT level at the time of diagnosis was an independent risk factor of mortality.
The study is novel and interesting. There are, however, some issues that need to be addressed.
There is compelling evidence that stroke outcome is strongly influenced by a variety of factors related to metabolic homeostasis, inflammatory response, perfusion disturbances, and drug actions (Ref. Neutrophil-to-Lymphocyte Ratio and Symptomatic Hemorrhagic Transformation in Ischemic Stroke Patients Undergoing Revascularization. Brain Sci 2020; Glycosylated Hemoglobin and Functional Outcome after Acute Ischemic Stroke; Early ficolin-1 is a sensitive prognostic marker for functional outcome in ischemic stroke). All these factors may act either at the brain site of damage and systemic level influencing the neurovascular recovery, secondary damage, and systemic complications. Accordingly, the ever-growing insight into stroke pathophysiology and course could aid to identify the clinical, biochemical or brain imaging parameters that may serve to improve prognostic algorithms.
Authors did not provide data on blood pressure (BP) and BP variability, but recent emerging lines of evidence suggested that high BP and BP variations are important predictor of the prognosis of acute stroke (Increased blood pressure variability after endovascular thrombectomy for acute stroke is associated with worse clinical outcome. J Neurointerv Surg. 2018; Blood Pressure Variability: A New Predicting Factor for Clinical Outcomes of Intracerebral Hemorrhage. J Stroke Cerebrovasc Dis. 2020). If these findings may be available, this information would be helpful in this study result. Otherwise, discuss and update the manuscript according to suggested evidence and highlight this issue among the study limits.
It is not clear how statistical analyses were performed. Cox proportional hazard analysis evaluates associations with survival time and not survival rate. How did you select the possible confounding factors to include into Cox proportional regression analysis? Did you select them on the basis of univariate comparison? At which p value threshold? Table 2 is not displayed in the manuscript.
Author Response
Responses to reviewers’ comments
We are very grateful for your valuable comments and suggestions. We have revised our manuscript according to your recommendations.
All revisions are marked in red in the text.
Reviewer A's comment #1.
There is compelling evidence that stroke outcome is strongly influenced by a variety of factors related to metabolic homeostasis, inflammatory response, perfusion disturbances, and drug actions (Ref. Neutrophil-to-Lymphocyte Ratio and Symptomatic Hemorrhagic Transformation in Ischemic Stroke Patients Undergoing Revascularization. Brain Sci 2020; Glycosylated Hemoglobin and Functional Outcome after Acute Ischemic Stroke; Early ficolin-1 is a sensitive prognostic marker for functional outcome in ischemic stroke). All these factors may act either at the brain site of damage and systemic level influencing the neurovascular recovery, secondary damage, and systemic complications. Accordingly, the ever-growing insight into stroke pathophysiology and course could aid to identify the clinical, biochemical or brain imaging parameters that may serve to improve prognostic algorithms.
Answer)
> Thank you for the suggestion. As you pointed out, there are evidences for metabolic factors and inflammatory factors as prognostic factors for ischemic stroke. Therefore, in this study, ESR was used as an inflammatory factor, but an additional literature review on NLR was conducted, and it was determined that it would be better to add it to the study, so we included it. There was a tendency of association with the prognosis, but there was no long term effect. In addition, glucose level at admission was used in this study as a metabolic factor. We also reviewed the HbA1c and ficolin recommended by the reviewer and tried to include it in the study, but we could not include it in the study because there were many patients who were not tested at the time of diagnosis of stroke.
For the additionally investigated NLR, it was added to the method, the following paragraph was added to the discussion line, and the limitation was additionally described as follows:.
Discussion
….
There is strong evidence that stroke outcomes are strongly influenced by various fac-tors related to metabolic homogeneity, inflammatory reactions, perfusion disorders [32-35]. It is known that glycosylated hemoglobin or serum glucose levels at admission are associated with metabolic homeostasis in the prognosis of patients with acute ischemic stroke. In this study, there were many missing data on glycosylated hemoglobin due to the characteristics of retrospective studies, so glucose at admission was used as a metabolic factor. However, in this study, the long-term effect of glucose levels on mortality was not significant.
…..
In addition, as previously mentioned, ficolin-1, which is associated with inflammatory reactions, was not a test widely distributed in general clinical sites, and thus could not be investigated due to limitations in retrospective design. Although a clear mechanism has not been identified yet, if a prospective design study including compelling evidence relat-ed to the prognosis of stroke is conducted, it will be helpful to improve the algorithm for the prognosis of ischemic cerebral infarction in the future
Reviewer A's comment #2
Authors did not provide data on blood pressure (BP) and BP variability, but recent emerging lines of evidence suggested that high BP and BP variations are important predictor of the prognosis of acute stroke (Increased blood pressure variability after endovascular thrombectomy for acute stroke is associated with worse clinical outcome. J Neurointerv Surg. 2018; Blood Pressure Variability: A New Predicting Factor for Clinical Outcomes of Intracerebral Hemorrhage. J Stroke Cerebrovasc Dis. 2020). If these findings may be available, this information would be helpful in this study result. Otherwise, discuss and update the manuscript according to suggested evidence and highlight this issue among the study limits.
Answer)
> As you pointed out, there are evidences for BP and BPV as prognostic factors for ischemic stroke. Therefore, during a literature review, a meta-analysis paper was reviewed that among BP and BPV, systolic had the most significant association. Accordingly, it was concluded that SBP and SBPV were added to the study and included through further investigation. There was a tendency of association with the prognosis, but there was no long term effect.
The additionally investigated SBP and SBPV were added to the method, and the following paragraphs were additionally described in the discussion line.
Discussion
…..
Recent evidence suggests that high blood pressure and blood pressure variability are important factors in predicting the prognosis of acute stroke. In particular, it is known that systolic blood pressure at admission and systolic blood pressure variability increase the risk of cerebral hemorrhage or symptomatic hemorrhagic transformation and adversely affect mortality and functional outcome [30-32]. As a result of our study, in the comparison between the expired group and the surviving group, there was no difference in SBP at admission, but in the case of SBPV, there was a significant difference between the two groups. However, there was no statistically significant difference when observing the long term effect. This is thought to be because majority of the subjects included in this study did not progress to cerebral hemorrhage or hemorrhagic transformation.
…..
Reviewer A's comment #3.
It is not clear how statistical analyses were performed. Cox proportional hazard analysis evaluates associations with survival time and not survival rate. How did you select the possible confounding factors to include into Cox proportional regression analysis? Did you select them on the basis of univariate comparison? At which p value threshold? Table 2 is not displayed in the manuscript.
Answer)
The Cox regression used in this study used the Forward stepwise: Conditional method. It is a method similar to the stepwise method of linear regression test, in which the program automatically enters statistically significant factors. As you pointed out, there is no detailed description of the statistical method, so we have described it in detail in the method section. It also describes in detail what factors are included, and additionally describes them in the table.
Methods
….
(Forward stepwise: Conditional) that controlled for possible confounding factors. The covariates included in the multivariate Cox proportional regression analysis (Forward stepwise: Conditional) were as follows: age, sex, TOAST classification, stroke location, hemispheric localization, initial NIHSS at admission, SBP at admission, SBPV, thrombolytic treatment, significantly reduced ALT level (less than 10 U/L), reduced albumin level (less than 3.5 g/dL), reduced Hb level (less than 11 g/dL), abnormal glucose level (more than 7.3 or less than 3.7 mmol/L), serum Cr, ESR, NLR ,total cholesterol levels, BMI, and history of hypertension, diabetes mellitus, atrial fibrillation, coronary artery occlusive disease, cancer, smoking, and alcohol consumption.
Reviewer 2 Report
I would like to thank the editor for giving me the opportunity to review the manuscript entitled "Significantly Reduced Alanine Aminotransferase Level Increases All-Cause Mortality Rate in the Elderly after Ischemic Stroke". Stroke is a leading cause of death worldwide and increased disability, and represents one of the major health burdens. Understanding the factors predicting a good prognosis is mandatory both to optimize choices of treatment, follow-up monitoring, and eventually identify modifiable risk factors (e.g., diet, physical activity, etc.). Frailty is a multifactorial condition characterizing a large proportion of the elderly, and is know to predict bad outcomes. I find these results about ALT levels intriguing and may add a value to the current knowledge of prognostic biomarkers.
However, I have some suggestions to improve the manuscript and to give a better clinical understanding:
- In the methods section, authors refer to NIHSS. It is not clear to me if they mean the NIHSS at admission, or the NIHSS at discharge. Addendum: later I read that in the statistics part you write "initial NIHSS". Please, clearify it before when you introduce the NIHSS. Eventually, you could also present the discharge NIHSS.
- Another clinical score that well predicts function outcomes, and might be associated to frailty, is the modified Rankins Score (mRS). Can the authors, if present, add this score (anamnestic and/or at discharge) to their analysis?
- I could not find in the manuscript any detail about the principal therapy the patients received upon admission (i.e., pharmacological or mechanical reperfusion with tPA agents or EVT). This, together with the time to treatment information, can better describe the sample as they might strongly explain the increased mortality (see, as example, Ajcevic et al., Physiol Meas, 2020).
Author Response
Responses to reviewers’ comments
We are very grateful for your valuable comments and suggestions. We have revised our manuscript according to your recommendations.
All revisions are marked in red in the text.
Reviewer B's comment #1
- In the methods section, authors refer to NIHSS. It is not clear to me if they mean the NIHSS at admission, or the NIHSS at discharge. Addendum: later I read that in the statistics part you write "initial NIHSS". Please, clearify it before when you introduce the NIHSS. Eventually, you could also present the discharge NIHSS.
Reviewer B's comment #2.
- Another clinical score that well predicts function outcomes, and might be associated to frailty, is the modified Rankins Score (mRS). Can the authors, if present, add this score (anamnestic and/or at discharge) to their analysis?
Answer)
The initial NIHSS stands for NIHSS at the time of admission. As pointed out, it may cause confusion of meaning, so the text and table were changed to initial NIHSS at admission. (The mRS and NIHSS at the time of discharge was not able to be added due to the large number of missing data. This is mentioned in the limitation.)
Discussion
….
Since this study was a retrospective study, information on the modified rankin scale or NIHSS at discharge was difficult to control and was not included.
…..
Reviewer B's comment #3
- I could not find in the manuscript any detail about the principal therapy the patients received upon admission (i.e., pharmacological or mechanical reperfusion with tPA agents or EVT). This, together with the time to treatment information, can better describe the sample as they might strongly explain the increased mortality (see, as example, Ajcevic et al., Physiol Meas, 2020).
Answer)
As pointed out, an additional investigation was conducted on the thrombolytic treatment, and it was added to the method, result, and table. However, only whether or not thrombolytic treatment was included in the statistical model, but there were no significant results. Subgroup analysis for time to treatment was not possible. (Since the sample size was small, subgroup analysis was not possible, and this was additionally described in the discussion and limitations.)
Data Collection
Age, sex, and body mass index (BMI) data were evaluated. The investigated medical data related to ischemic stroke included lesions of ischemic stroke, thrombolytic treatment (including IVtPA [Intravenous Tissue Plasminogen Activator] and stent retriever), and etiology as per the Trial of ORG 10172 in Acute Stroke Treatment (TOAST) classification and history of atrial fibrillation, diabetes, hypertension, coronary artery occlusive disease, cancer, alcohol, and smoking.
Discussion
…
Thrombolytic treatment includes IVtPA treatment and stent retriever treatment. Only 9.3% of the subjects included in this study were included. It is known that the survival rate of thrombolytic treatment may vary depending on the treatment time, but in this study, the sample size is small for subgroup analysis, so it cannot be concluded.[28, 29]
….
In addition, the limitation of this study is that the subgroup analysis for thrombolytic treatment was not progressed due to the small sample size.
….
Round 2
Reviewer 1 Report
The Authors addressed all the issue. Please, correct the typo "ficloin".
Author Response
We are very grateful for your one comment. We have revised our manuscript according to your recommendations.
One-point revisions are marked in red in the text.
Thank you very much for your kind review.
Best regards,
Sang Joon An, MD
Assistant Professor
Department of Neurology
Catholic Kwandong University of Korea College of Medicine
International ST. Mary`s Hospital
Simgok-RO 100GIL 25, Seo-GU, Incheon Metropolitan City, 22711, Republic of Korea
Email: neuroan@gmail.com
Mobile: +82-10-5123-5990
Reviewer 2 Report
I thank the authors for the time spent revising the manuscript according to the suggested comments. I recommend to accept the present manuscript and congratulate with the authors for the nice study and the extensive revision.
Author Response
It was an honor to be advised last your recommendations.
Thank you very much for your kind review.
Best regards,
Sang Joon An, MD
Assistant Professor
Department of Neurology
Catholic Kwandong University of Korea College of Medicine
International ST. Mary`s Hospital
Simgok-RO 100GIL 25, Seo-GU, Incheon Metropolitan City, 22711, Republic of Korea
Email: neuroan@gmail.com
Mobile: +82-10-5123-5990